# Functional ultrasound imaging of the brain reveals propagation of task-related brain activity in behaving primates

Alexandre Dizeux[1], Marc Gesnik [1], Harry Ahnine[2], Kevin Blaize[3], Fabrice Arcizet [3], Serge Picaud [3], José-Alain Sahel[3,4], Thomas Deffieux[1], Pierre Pouget[2] & Mickael Tanter[1]

Neuroimaging modalities such as MRI and EEG are able to record from the whole brain, but this comes at the price of either limited spatiotemporal resolution or limited sensitivity. Here, we show that functional ultrasound imaging (fUS) of the brain is able to assess local changes in cerebral blood volume during cognitive tasks, with sufficient temporal resolution to measure the directional propagation of signals. In two macaques, we observed an abrupt transient change in supplementary eye field (SEF) activity when animals were required to modify their behaviour associated with a change of saccade tasks. SEF activation could be observed in a single trial, without averaging. Simultaneous imaging of anterior cingulate cortex and SEF revealed a time delay in the directional functional connectivity of 0.27 ± 0.07 s and 0.9 ± 0.2 s for both animals. Cerebral hemodynamics of large brain areas can be measured at high spatiotemporal resolution using fUS.

[1] Physics for Medicine, ESPCI, INSERM, CNRS, PSL Research University, Paris, France. [2] INSERM 1127, CNRS 7225, Institut du Cerveau et de la Moelle épinière, Sorbonne Université, Paris, France. [3] INSERM, CNRS, Institut de la Vision, Sorbonne Université, Paris, France. [4] Department of Ophthalmology, The University of Pittsburgh School of Medicine, Pittsburgh, PA, USA. These authors jointly supervised this work: Pierre Pouget, Mickael Tanter. Correspondence and requests for materials should be addressed to A.D. (email: alexandre.dizeux@gmail.com) or to P.P. (email: pierre.pouget@upmc.fr) or to M.T. (email: mickael.tanter@gmail.com)

Many if not all neuroscientific techniques measuring brain activity present with a compromise between the size of the imaging field, temporal and spatial specificity, sensitivity, and physical constraints on the animal. Electrophysiological, and more recently, two-photon microscopy techniques, now permit the recording of neuronal activity at high sampling rates; however, their field of view is limited by the number of implantable electrodes or light scattering in tissues[1,2]. Several other imaging techniques can be used to detect changes in metabolic activities following neural activities, including perfusion fMRI and contrast fMRI. Blood oxygen level-dependent BOLD contrast imaging fMRI is currently the most widely used fMRI method. Although BOLD fMRI provides only an indirect measure of neuronal activity, there is strong empirical evidence that the BOLD signals are indeed highly correlated with neuronal activities[3]. The signal in perfusion fMRI has also been described as more stable compared to BOLD responses[4], and leading to better spatial specificity. However, these methods require long acquisition sequences, which limit their use in human or awake behaving animals, and they are less sensitive than BOLD. The relatively weak signal-to-noise ratio of these techniques also remains a serious constraint. This means that multiple trials/ conditions must be averaged to statistically analyze the changes in activity, and real-time modulations cannot be unequivocally associated with a given brain region. The latest functional ultrasound imaging techniques (fUS), based on ultrafast Doppler, offer a different way to monitor brain hemodynamics[5–7] and functional connectivity[7]. This technique has been successfully applied in rodents to measure CBV with high spatiotemporal resolution and sensitivity, but had never been used in non-human primates.

Another approach is to use ultrafast Doppler (neuro-functional ultrasound), whose signal is directly proportional to the number of moving red blood cells (RBCs) in the sample volume, in other words to the local blood volume[8]. Ultrafast Doppler measures real-time CBV changes down to a typical 5% increase while CBV changes assessed by ultrafast ultrasound typically reach more than 50% increase during stimulus-based or brain endogenous activity in small vessels, resulting in a very high sensitivity. This increase in CBV is present in arterioles as well as in capillaries. The traditional view that CBF is regulated solely by precapillary arterioles has recently been challenged by studies in retinal and cerebellar slices[9]. Kleinfeld and colleagues showed that both the average velocity and density of RBCs are greater at high values of flux than at low values using two-photon laser scanning microscopy to image the motion of RBCs in individual capillaries below the pia mater of the primary somatosensory cortex in rats[10]. Although fUS is not able to resolve individual capillaries, it detects tiny blood flow changes (down to 0.5 mm/s blood flow speed) in individual pixels. Miniaturization of ultrafast ultrasound technology has recently made it possible to extend this whole-brain neuroimaging to awake and freely moving rats[6]. However, the instantaneous monitoring of endogenous brain signals during cognition has never been demonstrated, though the interest of using this modality to explore brain network dynamics is clear. The combination of high sensitivity, high spatiotemporal resolution (typically 100 μm and 10 ms), and large field of view (ranging from cm² to tens of cm²) that fUS offers is key to the dynamic study of endogenous brain signals and patterns during visual tasks that we present here.

In the present study, we provide the first fUS images captured from awake and behaving non-human primates performing

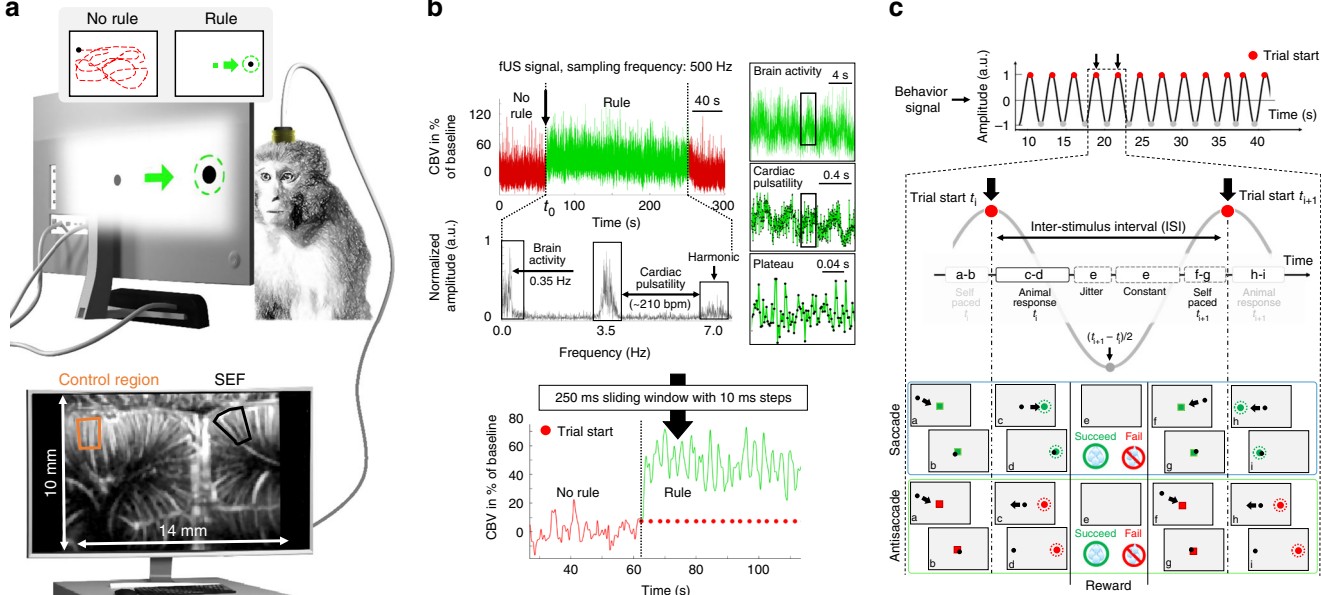

**Fig. 1** Setup and signal information. **a** The high sensitivity to cerebral blood volume (CBV) changes in fUS (functional ultrasound) imaging enable the single trial detection of supplementary eye field (SEF) activation during visual tasks. The animals performed in a row with baseline (rest phase), fixation, saccade, and antisaccade trials while CBV changes were recorded in fUS with a FOV of 10 × 14 mm. **b** fUS signal was recorded at a sampling frequency of 500 Hz the SEF region; below is the associated spectrum (fast Fourier transform) of the SEF signal during the visual task. Peaks frequency observed at 0.35 and 3.5 Hz are related to brain activity and cardiac pulsatility (~ 210 bpm), respectively (7 Hz is a harmonic frequency of cardiac pulsatility). To obtain a cleaner signal, cardiac pulsatility was first removed with a cutoff filter, and then a 250 ms sliding window with a time increment of 10 ms was applied in each pixel of the image. **c** Behavior signal consists of a sinusoid for which each period was defined by the time between trial start $t_i$ and the following trial start $t_{i+1}$. Jitter and constant time were similar for all type of task; the only significant variation between visual tasks was the animal response time (RT). Each visual task was initialized by the animal (**a, b**), for saccade he has to hit the cue (**c, d**) whereas for antisaccade he has to hit the opposite side where the cue appeared (**c, d**) and finally for the fixation task (not represented in the figure) he has to keep its eye's position on the initial cue. Depending on the result, the animal received a reward (**e**) and he then could initialize a new trial sequence (**f–i**)

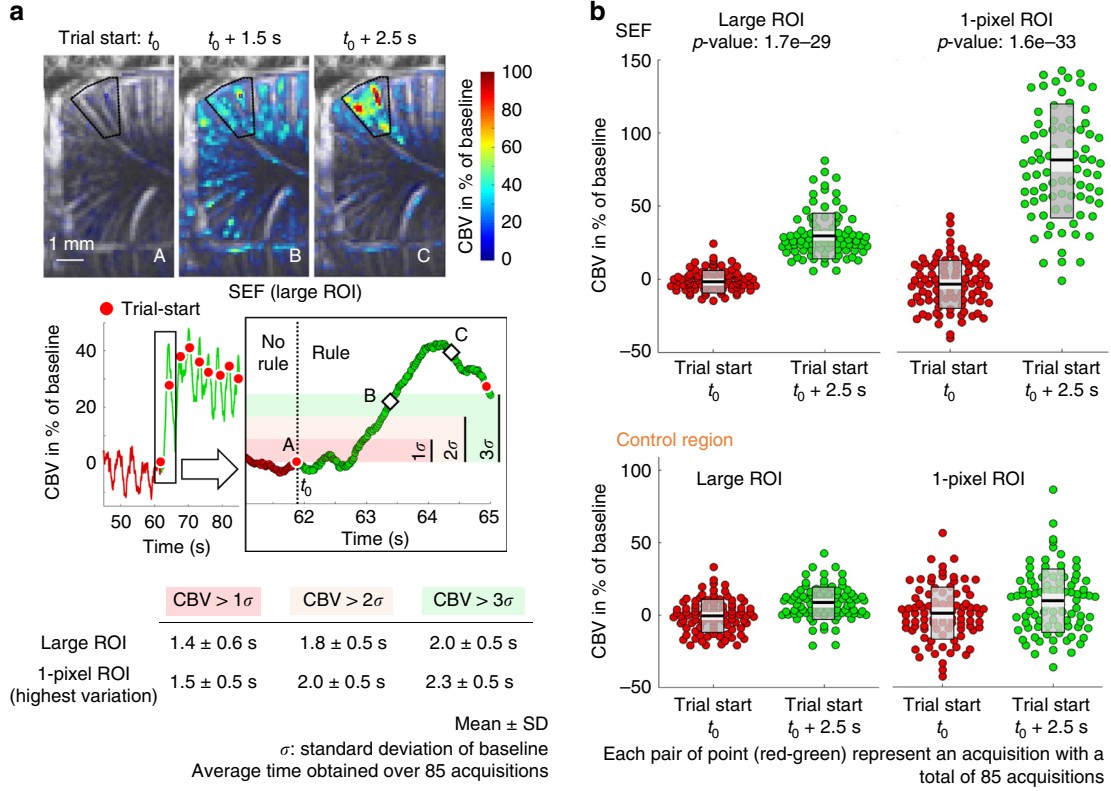

**Fig. 2** fUS detect a rule change from the single first trial. **a** The start of the first visual cue induced a strong CBV variation of ~40% in the SEF reached in 2.5 s (quantified in a large ROI delineated by a black line, ~240 pixels), and the CBV was already twofold greater than the standard deviation of baseline ($\sigma$) after 1.5 s. Whether CBV was quantified in a large ROI or in a 1-pixel ROI (pixel chosen with the highest variation of CBV over the experiment), the delay to observe CBV above a certain range of $\sigma$ was similar (average over 85 acquisitions). **b** The CBV variation between the beginning of the first trial and 2.5 s after was highly significant in the SEF (in the large and 1-pixel ROI), whereas no difference was observed in the control region (paired-sample $t$-test). Raw data (red and green circles) were summarized with mean (solid black line), 95% confidence interval (light gray area), and one standard deviation (transparent gray area)

complex tasks. The supplementary eye field (SEF) is an area of the dorso-medial frontal cortex active during eye movement tasks[11,12]. Evidence from neural recording and stimulation indicate a role for the SEF in learning arbitrary oculomotor stimulus–response rules[13], reward or error monitoring[14–16], encoding object-centered directions for saccades[17], smooth pursuit[18,19], self-paced eye movements[20], unpredictable sequential eye movements[21], antisaccades[22], and the execution of memory-guided saccade sequences[23]. In humans, only a few studies have attempted to determine the function of the SEF by examining failures in saccadic performance of patients with lesions subsuming this area[24]. The neurons of the anterior cingulate cortex (ACC) in macaque monkeys discharge in response to visual stimuli and during the following saccadic eye movements[25]. Functional imaging in humans has described activation in the posterior cingulate cortex to be associated with visually guided saccades[26]. Other evidence indicates a role in gaze control for a caudal zone in the anterior cingulate cortex[27]. Saccadic eye movements can be evoked by electrical microstimulation of a region in the upper bank of the cingulate sulcus directly ventral to the SEF, in area 24c[28]. Furthermore, functional brain imaging studies have also reported activation in the anterior cingulate cortex during the production of self-generated saccades guided by arbitrary cues[29]. In this study, we simultaneously recorded fUS images from the SEF and ACC regions in macaques freely performing cognitive tasks. CBV changes induced by the neuronal activity during a single task were recorded, without requiring any statistical averaging over several task repetitions as is the case in

fMRI or electrophysiology. We show the possibility of recording cerebral hemodynamics at a high spatiotemporal resolution (100 μm, 10 ms) and sensitivity with simultaneous monitoring of task-related performance through eye tracking. We first present the details of this experimental framework and ability of fUS imaging to precisely image no task/task transitions (Figs. 1, 2). We then demonstrate that fluctuations in CBV changes recorded by fUS during low and high oculomotor control tasks are temporally synchronized with the individual trials (Fig. 3), and we demonstrate that this level of synchronization is correlated with, and even predictive of, the success rate. Finally, we indicate the ability of neuro-functional ultrasound to detect the dynamic propagation of local CBV changes through cortical layers and between the SEF and ACC (Fig. 4).

## Results

**fUS reveals brain activation related to cognitive tasks**. We first imaged cue-evoked hemodynamic responses in the SEF of awake behaving primates as they were instructed to start an oculomotor task. Twenty successive trials (fixation, saccade, and antisaccade trials) were performed. We observed the responses using fUS imaging with a 15 MHz ultrasonic array (see Fig. 1a) inserted into the $20 \times 20\ mm^2$ electrophysiological chamber. The high temporal sampling rate (10 ms) enabled by fUS imaging means that cardiac pulsatility and respiratory motion artifacts occurring at different Doppler frequencies could be unambiguously subtracted from the collected data (Fig. 1b, see Supplementary Fig. 1a for acquisition reproducibility). Correlation maps over all trials

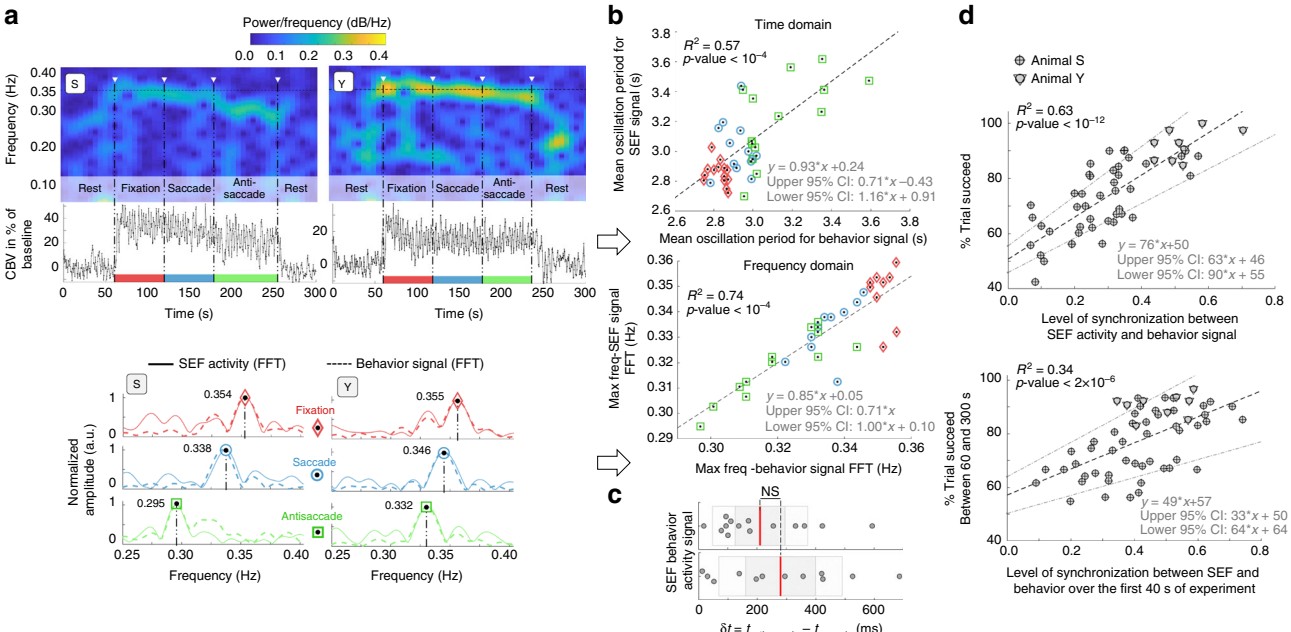

**Fig. 3** Relationship between CBV oscillations, behavior and success rate. fUS imaging is able to differentiate the task response times for fixation/saccade and antisaccade using frequency analysis of CBV changes in the SEF as well as the success rate. Panel **a** presents an example of two spectrograms related to CBV oscillations in the SEF (large ROI, ~240 pixels) for both animals (S and Y). During the visual task, the CBV in the SEF oscillates around 0.35 Hz and decreases during the rest phase. For each visual task, the fast Fourier transform of the temporal signal in the SEF and the related behavioral signal were plotted and superimposed. A match in peak frequency between the SEF signal and behavior signal indicates synchronization between brain activity and behavior. The behavior signal was created by plotting a sinusoid for which each period was defined by the delay between trial start $t_i$ and $t_{i+1}$, and as shown in Fig. 1c, this signal contains the response time of animals. **b** Whether in the time domain or frequency domain, there was a set of 13 acquisitions for which a significant correlation between CBV oscillations in the SEF and a related behavior signal was highlighted. **c** fUS imaging permitted the assessment of the subtle changes in task response time between the saccade and antisaccade experiments ($\delta t = t_{antisacc} - t_{sacc} = 280 \pm 210$ ms) in agreement with behavior data ($\delta t = 210 \pm 160$ ms, no significant difference, $p = 0.36$) using a limited set of 20 successive trials, raw data (gray circles) were summarized with mean (solid red line), 95% confidence interval (gray area), and one standard deviation (white area). **d** The correlation (level of synchronization) between the SEF signal and behavior signal has been plotted for all acquisitions and was significantly correlated to the success rate of the animal. It was even possible to predict the success rate of the animal only by considering the brain-behavior synchronization over the first 40 s of the experiment. Coefficient correlation and $p$-value were computed using Pearson correlation test

between the no-rule and rule trials reveals a strong activation of the SEF region during the rule trials (Fig. 2a). The peak amplitude of the CBV signal increased in the SEF by $31.3\% \pm 15.2\%$ on average over baseline in a large region of interest (ROI, ~240 pixels) and $84.8\% \pm 38.7\%$ in a 1-pixel ROI in all 85 recorded sessions (similar results were obtained in ROI of 6 and 12 pixels, but for more clarity data were not shown). In the large and 1-pixel control regions, the average variation and the standard deviation were $8.8\% \pm 10.9\%$ and $8.7\% \pm 21.6\%$, respectively (Fig. 2b).

**fUS could detect a rule change from the single first trial.** The high sensitivity of the technique also allows fast CBV dynamics during rule change from the single first trial to be detected. CBV changes revealed a transient, abrupt, and localized activation of the SEF region at the transition between no-rule/rule conditions (Fig. 2a). We quantified this abrupt variation of activity within the first trial of each block condition to assess the ability of cognitive fUS to monitor instruction changes within a 3 s window (Fig. 2a). Within each condition block, we quantified the activity within a single trial to assess the sensitivity of fUS. Brain activation maps of the correlation between the CBV signal and appearance of the cue revealed strong and abrupt significant activation changes in the region of the SEF (Fig. 2a). The CBV signal rapidly reached its maximum after the onset of the cue and was above two standard deviations of the baseline just after $1.8 \pm 0.5$ s (mean ± SD, Fig. 2a).

**Oscillations in fUS signals capture animal response times.** Beyond correlating CBV changes on a block design basis (rule/ no-rule conditions), we exploited the temporal resolution and sensitivity of fUS to more finely analyze the CBV variations between individual trials in each fixation, saccade, and anti-saccade block. CBV changes within each block were found to fluctuate at an instantaneous frequency synchronized with the effective trial repetition frequency. The spectrogram of the CBV signal in the SEF revealed 0.354, 0.338, and 0.295 Hz repetition frequencies for fixation, saccade, and antisaccade tasks, respectively (Fig. 3a, see Supplementary Fig. 2a for control regions). This temporal evolution of CBV estimated for small blocks of 20 successive trials was found to be highly correlated with the average time between trials ($R^2 = 0.74$, $p < 10^{-4}$, Pearson correlation test, Fig. 3b) measured by the eye tracking system. We found fUS imaging could inherently capture small changes in task response time between the saccade and antisaccade experiments ($\delta t = t_{antisacc} - t_{sacc.} = 280 \pm 210$ ms for fUS versus $\delta t = 210 \pm 160$ ms, for average time and standard deviation between trials estimated by the eye tracking system with no significant difference between both approaches: $p = 0.36$, paired $t$-test, Fig. 3c). Furthermore, the degree of synchronization between the inter-trial timings and the CBV responses in the SEF was measured and found to be significantly correlated with the percentage of successful trials ($R^2 = 0.63$, $p < 10^{-12}$, Pearson correlation test, Fig. 3d). The estimation of this degree of synchronization during the first 40 s (i.e., during the fixation block) was found to predict

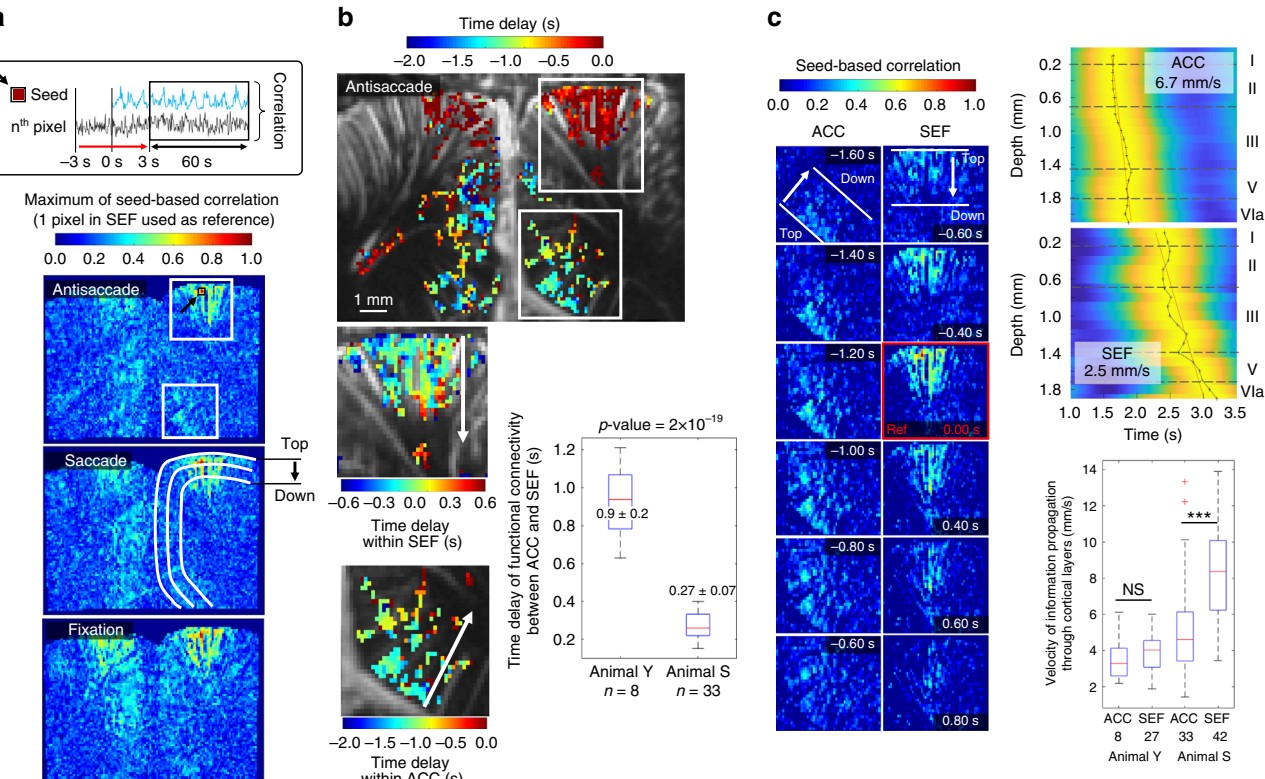

**Fig. 4** fUS permits access to the directional functional connectivity of cortical layers. **a** Time delays between and within the ACC and SEF were extracted from a seed-based correlation map for pixels with a correlation above 0.3 (a reference pixel in the SEF was chosen based on the highest variation in CBV over the experiment). **b** The average time delay of information propagation between the ACC and SEF was 1.0 ± 0.3 and 0.26 ± 0.08 s for animals Y and S, respectively. **c** Thanks to the temporal resolution of fUS (10 ms), it was possible to track the propagation of the correlated information within cortical layers and to compute the propagation velocity. Successive screenshots revealed a propagation of information first in the ACC and then in the SEF. The reference time (screenshot with red borders) corresponds to the time when the auto-correlation coefficient of the 1-pixel SEF is 1. The average velocity of information propagation through the cortical layers of the ACC and SEF was, respectively, 3.6 ± 1.3 and 4.0 ± 1.2 mm s⁻¹ for animal Y and 5.2 ± 2.6 and 8.4 ± 2.6 mm s⁻¹ for animal S. NS means non-significant and results are presented as mean ± SD. The data were summarized as boxplots depicting the minimum, lower quartile, mean value (red line), upper quartile, and maximum values. *** indicates a *p*-value <5 × 10⁻⁴ for a two-sample *t*-test. Digits on the *x*-axis of the boxplot indicate the number of measurements. Cortical layers were delineated based on reference[57]

the future percentage of successful trials ($R^2 = 0.34$, $p < 10^{-5}$, Pearson correlation test, Fig. 3d) between 60 and 300 s (i.e., during the saccade and antisaccade blocks). The synchronization during fixation could thus reveal the attention degree of the animal for a particular session.

**fUS tracks the directional functional connectivity**. We also studied the ability of fUS imaging to provide information on the directional propagation of SEF and ACC information, both within the cortical layers, and between cortical regions. Correlating the time profile of CBV variations in a seed pixel of SEF during the 20 successive trials with all other pixel time profiles revealed that functional connectivity between cortical regions (SEF and ACC) was similar for the fixation, saccade, and antisaccade experiments (Fig. 4a). Analyzing the time profile of the correlation signal versus depth in the cortical structures clearly revealed a top-down propagation of the maximal correlation values in the SEF layers (see Supplementary Movie 1) in animal Y during the fixation, saccade, and antisaccade tasks (Fig. 4b, see Supplementary Fig. 3a for animal S). Extracting these signals for pixels located at different depths within the ACC ($n = 8$) and SEF ($n = 33$) for animal Y exhibited precise propagation timing between upper and lower regions at a 4.0 ± 1.2 and 3.6 ± 1.3 mm/s speed (mean ± SD, Fig. 4c). Similar propagation speeds within the

ACC ($n = 33$) and SEF ($n = 42$) were found for animal S (5.2 ± 2.6 and 8.4 ± 2.6 mm/s , Fig. 4c).

An average time delay in the directional functional connectivity of 0.27 ± 0.07 and 0.9 ± 0.2 s was found between the ACC and SEF for animals S and Y (mean ± SD, Fig. 4a). These differences were not found to be correlated with success rate or other behavior parameters. For animal Y, some experiments revealed a directional functional connectivity time delay of 1.6 ± 0.2 s going from the SEF to ACC (see Supplementary Fig. 3b). The propagation of SEF-correlated information in one way or the other between the ACC and SEF were found in 30% (8/27) and 79% (33/42) of visual task blocs for animals Y and S, respectively (see Supplementary Fig. 3c for an example where there was no SEF-correlated information detected in the ACC).

## Discussion
In this study we highlighted the benefit of using fUS imaging to monitor activity of the prefrontal cortex (SEF and ACC) of non-human primates at rest or while performing visual tasks (fixation, saccade, and antisaccade). Based on event-related variation of CBV amplitude in the SEF, we were able to detect neuronal activity after only a single trial (Fig. 2a, b). The high sampling rate (10 ms) enabled us to show that CBV oscillation frequency in the SEF during block design sequences was related to the response time (RT) of the animals (Fig. 3b, c). Moreover,

the level of SEF-behavior synchronization revealed a strong correlation with success rate of animal and this level of synchronization over the first 40 s (part of the fixation task) was even predictive of future success rate (Fig. 3d). Finally, computation of lag from seed-based (SEF) correlation highlighted propagation of correlated information from ACC to SEF for both animals.

As shown in Fig. 2a, it was possible to follow a significant increase of CBV amplitude above the baseline level after a single trial. Due to the very high sensitivity of fUS imaging it was not necessary to average several trials to decrease the noise level. Most cognitive studies using fMRI measure BOLD variations related to task-evoked responses that usually involve averaging across many trials (block design) to improve confidence that BOLD variations are not artefacts. Other techniques can be implemented to lower signal from non-neuronal activity during acquisition by measuring and removing physiological parameters (respiratory and cardiac activity) from the BOLD signal through linear regression[30,31] or by designing an acquisition sequence with a higher sampling rate to avoid aliasing of a higher-frequency physiology[32,33]. This can be even done during post-processing using algorithms like independent component analysis[34] (ICA) or by regressing out signals that are common to all voxels[35]. Nevertheless, although event-related approaches in fMRI opened new areas of research in cognitive psychology, complex signal post-processing is still required to counter the limited spatiotemporal resolution and sensitivity of fMRI. Such approaches have been criticized for "[embedding] fMRI analyses within layers of abstraction—pulling researchers ever farther away from their data" as explained by Huette[36].

As presented in the spectrograms of Fig. 3a, a significant shift of the CBV oscillations frequency was found when changing rule of visual task. We verified the hypothesis (Fig. 3a, b) that the shift was directly related to the RT of animals by comparing the peak frequency of CBV signal spectra in SEF with the behavior signal spectra for each type of visual task (see Methods and Fig. 1c for detailed descriptions about behavior signal). It could be argued that this shift in frequency of the CBV signal in SEF could be related to fatigue of the animal instead of its response time. But as presented in supplementary figure 2b, we have performed several acquisitions with only baseline followed directly by antisaccade and we saw that CBV oscillations frequency were around 0.3 Hz for the whole acquisition which is typical frequency related to RT for antisaccade (Fig. 3b). In future work, with new primates, we will invert the order of visual tasks and we will increase the jitter between two start-trial.

The fact that fUS imaging does not require the implementation of any complex signal processing to extract the signal of interest from noise, even in an event-related study in a behaving nonhuman primate, is an important feature. Indeed, as summarized in supplementary figure 4, the noise level measured by fUS in regions outside of the brain had an amplitude variation of only 4%, ten times lower than the spontaneous coherent CBV fluctuations (40% of variation in amplitude) measured in a cortical region not involved in visual tasks, and 20 times lower than the CBV variations (80% variation from baseline to task) measured in the SEF region. As with event-related study, the recording of spontaneous brain activity in fMRI is challenging because the BOLD signal can be corrupted by, or even caused by, artefacts such as non-neuronal physiological fluctuations. Despite the fact that we did not investigate the relationship between low and high frequency of CBV oscillations in depth, fUS imaging was sensitive enough (Supplementary Fig. 4) to highlight spontaneous coherent CBV oscillations in behaving animals. Again, as presented in the example of supplementary figure 4, there is a ten-fold difference in amplitude between the noise signal and spontaneous CBV oscillations, and a two-fold difference between the latter and SEF activity.

Finally, fUS imaging was shown to be able to track directional functional connectivity in real time. Such directional functional connectivity has long been sought for by using fMRI. Mitra et al. examined the latency structure of spontaneous fluctuations in the fMRI BOLD signal[37]. They revealed that intrinsic activity propagates through and across regions on a timescale of ~0.5 s. They found variations in the latency structure of this activity resulting from sensory state manipulation (eyes open vs. closed), antecedent motor task (button press) performance, and time of day (morning vs. evening) clearly suggesting that BOLD signal lags reflect neuronal processes rather than hemodynamic delay. Their results emphasize the importance of the temporal structure of brain's spontaneous activity. In the quest for the dynamic activation in different cortical layers (laminae), the ultimate challenge of such laminar directional fMRI is to provide information on the direction of information flow by comparing the relative contributions of different laminae to the signal within a given patch of cortex. However, the rapid signal transmission across the neighboring laminae is an order of magnitude faster than that which fMRI can measure, which potentially jeopardizes the entire endeavor of laminar fMRI[38]. Bypassing this temporal resolution problem, Huber et al. recently proposed another strategy to provide evidence for laminar fMRI using a CBV-weighted fMRI approach and different stimuli paradigms[39]. In this study, we demonstrate that fUS can exploit its superior (by an order of magnitude) temporal resolution to track delays in the directional functional connectivity between the ACC and SEF. Within the SEF region, the propagation of seed-based correlated information was found between different layers, leading to a typical $210 \pm 70$ and $440 \pm 150$ ms (mean $\pm$ SD) delay between top and lower cortical layers (considering a distance of 1.6 mm), emphasizing the actual limit of fMRI for capturing this directional propagation of information. As timescales of the lag structure are of the order of seconds during task-evoked acquisitions, it could be argued that this is a purely vascular effect, due to conducted or retrograde dilation within blood vessels[40]. However, it is highly plausible that the contribution to the lag structure is here primarily neuronal for several reasons.

First, the propagation of a vascular wave due to vessel vasomotion would completely call into question the ability of fUS imaging to perform local measurements of brain activation. This would contradict the many studies where fUS imaging was reported to detect a localized activation in cortical or deep regions in rats[41,42], ferrets[43], and humans[44]. In particular, Bimbard et al. recently reported the ability of fUS to reconstruct tonotopic maps of cortical and deep structures, such as the inferior colliculus, with a resolution of 100 microns. During auditory stimuli, Bimbard et al reported the ability of fUS imaging to discriminate between the responsiveness of neighboring voxels in ferrets, with a functional resolution as fine as 100 μm. Furthermore, fUS imaging was found able to discriminate voxels based on their tuning curves within a distance of 300 μm in as little as 10 repetitions per frequency. Importantly, they reported this measurement as a conservative measure of functional resolution, since it largely depends on the smoothness of the underlying functional organization itself (tonotopy) and of the number of trials. The propagation of a pure vascular wave through all cortical layers would contradict these previous studies on the spatial resolution of fUS.

Second, it may be objected that the observed lag structure is due to regional variations in the latency of neurovascular coupling[45–47]. Although not completely refutable, this hypothesis is not probable here. The lag structure and propagation speed in the same region of the same animal are markedly different for

spontaneous and task-evoked activity, and cannot be explained by regional heterogeneity of the neurovascular coupling. The latency delays observed in fUS imaging during spontaneous activity (hundreds of ms) are an order of magnitude faster than the latency delays measured during task-evoked activity (some seconds) as has already been observed in the literature[48]. Lag structure results observed in fMRI are generally confined to a range of 0.5 s whereas latency in task-evoked responses are on a timescale of the order of several seconds[48].

Third, and even more importantly, the fast propagation speed of spontaneous activity we have found here is in good agreement with other studies[49–52]. Mohajerani et al. demonstrated that a propagation speed (typically 0.2 m/s) can be measured during spontaneous activity using VSD calcium imaging in mice. Their results avoid the question of neurovascular coupling and confirm the neuronal contribution of BOLD measurements in rs-fMRI. Our results are obtained with a blood flow imaging method presenting a temporal resolution (~10 ms) comparable with VSD calcium imaging which tracks neuronal activity. Both approaches yield the same order of magnitude for the propagation speed ($0.43 \pm 0.26$ and $0.36 \pm 0.09$ m/s for both primates in fUS compared to 0.2 m/s in VSD calcium imaging[53,54] and 0.4–6.3 m/s during sleep[51,52]). Importantly, the latency trajectory observed within and between ACC/SEF regions cannot be anatomically explained just by a vascular component propagation. Suppl. Movie 1 clearly exhibits a top-down directional connectivity in the ACC followed by a top-down directional connectivity in the SEF. These results support the models that predict that the ACC is activated earlier during preparatory periods, whereas the PFC monitors for conflict during stimulus processing and response selection[55,56].

Raw spatial correlation map presented in Fig. 4b demonstrates pixel-level activation in ACC. Although an increase of spatial filter could be applied to further improve the smoothing of activation maps, preserving high spatial resolution could be crucial for several applications, in particular, when a precise guiding of electrodes is required doing electrophysiology measurements.

Although these arguments render a neuronal basis for latency structure rather plausible, our fUS data provide only indirect proof. Further studies involving multimodality acquisitions (EEG, biphoton, and fUS) should allow us to unambiguously confirm the physiological basis of latency structure. Such neuroimaging of directional functional connectivity at a high temporal resolution offers wide perspectives for whole-brain studies investigating information flow between brain regions. In the present study, the exploration of brain function with fUS imaging was restricted to cortical areas because of the use of a 15 MHz probe, but it is possible to use an ultrasonic probe with lower frequency to record CBV fluctuations in a full slice of non-human primate as shown in supplementary figure 4a. The high spatiotemporal specificity and sensitivity of fUS imaging means that it is well-suited for behaving NHP studies and the development of complex experimental paradigms with, for instance, near real-time feedback loops between region activations and the experimental paradigm. Although current fUS imaging technology has some drawbacks compared to MRI—it is currently only 2D and requires a craniotomy—its high sensitivity, resolution and, equally important, its high portability and compatibility with other experimental equipment including electrophysiology[6] make it a compelling tool for innovative and interactive multimodal approaches to behaving studies in the NHP neuroscience field[57].

## Methods

**Animal model and behavioral data**. All experiments were ethically approved by the French "Ministère de l'Education, de l'Enseignement Supérieur et de la Recherche" under the project references APAFIS #561_2015042717569705 and #6355-2016080911065046. Functional data were acquired from two captive-born macaques (Maccaca mulatta) S and Y trained to perform various kinds of visual task (Fig. 1c). In the saccade task, the animal has to fix its gaze on the cue object presented on the right or left side of the screen; in the antisaccade task, it has to fix its gaze on the opposite side from where the cue appeared, and finally for the fixation task (not represented in the figure) it must maintain a steady gaze on the initial cue. Each animal was performing sequentially with baseline (rest phase), fixation, pro-saccade, and antisaccade trials in a blocked design of 60 s each (~20 trials/task), and this block was repeated five times for each day of acquisition. During data acquisition, the eye position of the primate was monitored at 1 kHz with an infrared video eye tracker (Eyelink 1k, SR-Research), which enabled live control of the behavioral paradigm and the delivery of a reward based on the success or failure of a visual task. The designation 'behavior signal' refers to an oscillating signal created with a period equal to the time between the start of trials, as shown in Fig. 1c. This period contains information about the time response of the primate.

**Implant and probe for fUS imaging in awake behaving monkeys**. The head fixation system was a titanium headpost (Non-dental acrylic implant, Crist Instrument, MD, USA). After behavioral training of the animals, a craniotomy of 20 mm × 20 mm above the supplementary region was performed (Medio-Lateral: 0; Antero-Posterior: +26), and an electrophysiological recording chamber was implanted (CILUX chamber, Crist instrument). A custom miniature 15-MHz ultrasound probe (Fig. 1a) (128 elements, 15 MHz, $100 \times 100$ μm² of spatial resolution) with acoustic coupling gel was placed in the chamber. The acquired images had a pixel size of $100 \times 100$ μm, a slice thickness of 400 μm and a field of view (FOV) of 14 mm × 10 mm (see Supplementary Fig. 4a for different FOV-related to different probe frequency). The FOV of our miniature probe allowed imaging of superficial and deep cortical regions including the SEF and the ACC.

**Functional ultrasound (fUS)**. We measured CBV variations with an fUS sequence modified from a previously used Power Doppler sequence[53]. As fUS signals are proportional to CBV[5], we refer to the acquired images as 'CBV images'. Data were acquired by emitting continuous groups of 11 planar ultrasonic waves tilted at angles varying from −10 to 10 degrees using an ultrafast ultrasound research scanner (256 electronic channels, 60 MHz sampling rate). Ultrasonic echoes were summed to create a single compound image acquired every 2 ms. After spatial temporal filtering based on the singular value decomposition of these ultrasonic images[54] for fine discrimination between blood flow and tissue motion, final Doppler images were created by averaging 125 compound ultrasonic images over a sliding 250 ms window with 10 ms overlaps. At a frame rate of 100 Hz, each 300 s acquisition period generated a final sequence of 30,000 Doppler images.

**Data processing**. The SEF was spatially located by mapping activated pixels obtained from computing the normalized correlation coefficient between the local Power Doppler signal obtained from fUS and the temporal block pattern (no-rule vs. rule) of the visual stimulus. The power Doppler signal was then averaged within a large region of interest (ROI) (~240 pixels) and a 1-pixel ROI in the SEF and in similar ROIs in a contra-lateral controlled region (Fig. 1a). The reference pixel in the SEF was chosen based on the highest variation of CBV during the recording sequence. A behavior signal was generated to enable CBV fluctuations in the SEF area to be temporally correlated with the behavior of the monkeys. The behavior signal consisted of a sine wave whose period was defined by the delay between trial start $t_i$ and the following trial start $t_{i+1}$ (see Fig. 1c). The variability in inter-stimulus interval (ISI) from one type of task to another is mainly influenced by the animal response time (RT), as the jitter and the self-paced period remain similar.

The level of brain-behavior synchronization was assessed by computing the correlation of SEF activity with the behavior signal. A linear regression was plotted between the SEF activity (y-axis) and the behavior signal (x-axis) for each visual task block (Fig. 3b), either in the frequency domain (peak frequency) or time domain (mean CBV oscillation period). For all experiments, the correlation between the SEF activity and behavior was plotted against the related success rate (Fig. 3d).

Spectrograms were drawn using the inbuilt MatLab function with a Hamming window of 15% for the overall acquisition time and 80% for overlapped samples (Fig. 3a, Supplementary Fig. 2a–d) to study the shift frequency of CBV oscillations in regions of interest during visual tasks.

## Data availability

All data and software supporting the findings of this study are available from the corresponding authors upon reasonable request.

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

## Acknowledgements

This work was supported by the European Research Council SYNERGY Grant scheme (HELMHOLTZ, ERC Grant Agreement #610110) and the Program (*FP7/2007–2013*)/ERC grant agreement no. 339244-FUSIMAGINE.

## Author contributions

M.T., P.P., and T.D., conceived the study, M.G. and T.D. developed sequence acquisition, A.D. and H.A. acquired data. A.D. performed data processing. M.T., P.P., T.D., J-.A.S., S.P., and A.D. interpreted the results and M.T., P.P., and A.D. wrote the first draft of the manuscript with substantial contribution from T.D., M.G., F.A., and K.B. All authors edited and approved the final version of the manuscript.

## Additional information

**Competing interests:** T.D. and M.T. are co-founders and shareholders in the ICONEUS company. The remaining authors declare no competing interests.

