## [Peer Review File · Nature Communications]

Reviewers' comments:

Reviewer #1 (Remarks to the Author):

The paper contains high quality data and figures and may be of interest to the neuroimaging community. However, it is extremely poorly written and is not in an academic style. Many of the sentences are poorly constructed and do not make sense. It is unclear whether the aim of this paper is to assess the efficacy of the imaging modality used or to explore the role of the specific brain regions in the cognitive task employed. If the aim is the former, then the temporal and spatial resolution and its depth sensitivity should be quantified. If it is the latter, then the introduction should introduce the brain regions and their function. The behavioural paradigm used is not explained and as such, the data is difficult to interpret or follow. The discussion does not summarise the findings and contains an unclear description of many imaging techniques. Some examples of the writing style errors and lack of clarity are provided below but are by no means details of every instance. The paper should be completely rewritten in clear simple academic scientific English with the aims and objectives of the study made clear and explicit.

Abstract

Which species?

"Here we demonstrate that neurofunctional ultrasound can exploit its millisecond time resolution within a few hundreds of microns to establish the dynamic role of brain region activations."
This is not really a sentence. Also, contradicts itself a "few hundred microns", then claims to be able to image deeper layers?

Peculiar use of the term avant garde ?

"This paves the way to some avant-garde live imaging developments and create novel perspective for deeper understanding of brain network dynamics."

Intro

Line 6 – size of what?

14 valuable is a value judgement, they are all valuable - depends what you are trying to do?

23 – possibly true but what about event related fMRI

36 instantly capture – what does this mean?

11 experimental framework, what is a framework, better to write in more simple clear terms.

"Nicely fluctuate" not typical academic style

Discussion

The first paragraph is unclear, which technique are we using here? Which technique are you comparing it to? And which has 5 or 50% sensitivity

"Functional data were acquired with 2 trained captive-born macaques (Maccaca 18 mulatta) performing in a row with baseline (rest phase),"

Were the 2 monkeys sat in a row as (as this sentence implies) or were the cognitive tasks performed sequentially?

Reviewer #2 (Remarks to the Author):

This study reports on a ultrafast Doppler-based imaging approach (fUS) for measuring fluctuations in cerebral blood volume (CBV) with a high spatial (100um) and temporal (10ms) resolution in two

well-separated brain areas (SEF and AC) of two awake, behaving non-human primates. While this is without a doubt a technical 'tour de force', I'm not sure what the main contribution is and question the biological relevance of the acquired data.

Major:

1- The authors make the point that fast imaging of CBV may help making direction inference of neural communication. However, CBV changes must be met with changes in vessel size (i.e. dilation/constriction) which typically takes place on the order of seconds (not ms, like neuronal activity). To me, Figures 1c, 3 and 6 seem to show this: Area AC responds ~1-2s before area SEF. This is well-beyond neuronal communication timescale, and more likely represents conducted or retrograde dilation within blood vessels. That is, dilation-mediated increases in CBV are first manifested in deep paryenchymal vessels (area ACC?) and then back propagate towards the larger surface vessels (area SEF?, also see O'Herron et al., Nature 2016). Should this be the case, it would suggest that they are not measuring neuronal communication between areas per se (i.e. a top-down neuronal effect), but rather simply providing further evidence for vascular communication, thus questioning the novelty of the current study. The authors need to address this.

2- I struggled with the meaning and significance of the 'behavioral signal'. A schematic showing the temporal evolution of a successful vs unsuccessful trial would have been helpful. As a result, I could not appreciate the relevance of the CBV-behaviour correlations.

Minor:

1. How was area AC defined?
2. Cardiac and respiratory artefacts should be validated with additional measures.
3. Figure 2a is impressive, though I'm surprised the authors did not exploit the data further by examining the spatial extent of this effect (rather than using a 240 pixel ROI, which seemed arbitrary to me).
4. The paragraph in the discussion describing the many limitations of BOLD/ASL is long and doesn't really add much since these data were not acquired in the current study, making it impossible to quantify the true SNR advantage of fUS. In my opinion, this would have substantially increased the importance of this work.
5. While it's true that BOLD may not be able to accurately measure rapid signal transmission across the laminae, BOLD is in part sensitive to O₂ metabolism, which is arguably a closer correlate to neuronal activity than CBF and CBV. In this sense, it could be argued that fUS suffers from inherent limitations (i.e. vessel dilations which are slow) thus making it no better than BOLD. The authors should comment on this.

Reviewer #3 (Remarks to the Author):

This paper examines the measurement of cerebral blood volume changes with function ultrasound (fUS) in response to visual cues in macaque primates. The blood flow in the supplementary eye field (SEF) and anterior cingulate cortex (ACC) was measured after different cues (fixation, saccade, and anti-saccade). The paper has interesting data, but needs to clarify several aspects. The temporal resolution of the fUS method is a strong advantage of this paper for resolving aspects of responses in the brain to visual cues.

1. There are a fair number of superlatives that appear throughout the text (Auspiciously) that may need to be toned down.
2. What is the role or point of the 1 pixel region-of-interest? This is not properly motivated and honestly shows much larger variation than using a larger area. This is expected, but the

justification of this alternative analysis is missing.

3. Figure 2. What defines trial success? This is a fundamental component that was not properly defined.

4. How was the seed for the seed-based correlation analysis chosen? Was the seed varied and did it affect the results?

5. In the SEF region the propagation is top-to-down, but in the ACC it is down-to-top. This needs to be reflected in the analysis and description.

6. The invasiveness of this technique needs to be addressed as a limitation in the Discussion.

7. The ethical approval for animal testing needs to be addressed in the Methods.

8. Figure 2d. What is the implication for the correlation for the first 40 seconds to later measurements?

9. Figure 3b. The points are very sparsely distributed which makes the results somewhat suspect with respect to the robustness. The authors need to address this. Would longer averaging times in slow time be useful to make more robust measurements?

10. Figure 3c. The Roman numerals on the right of the propagation plots need to be defined.

11. Figure 4. The "SC ?" need to remove the question marks.

12. Figure 6. Again, these correlation maps are incredibly sparse. How was the path of propagation (the spatial axis) defined? Was it done on a Cartesian grid as shown in the image or some other path? The path needs to be shown on the images. Also, in the estimation of the propagation the feature being tracked needs to be stated. There is a good deal of variation in the propagation plots and the R^2 values of the fits should be reported to provide a confidence metric for the reader.

Reviewers' comments:

Reviewer #1 (Remarks to the Author):

The paper contains high quality data and figures and may be of interest to the neuroimaging community. However, it is extremely poorly written and is not in an academic style. Many of the sentences are poorly constructed and do not make sense. It is unclear whether the aim of this paper is to assess the efficacy of the imaging modality used or to explore the role of the specific brain regions in the cognitive task employed. If the aim is the former, then the temporal and spatial resolution and its depth sensitivity should be quantified. If it is the latter, then the introduction should introduce the brain regions and their function. The behavioural paradigm used is not explained and as such, the data is difficult to interpret or follow. The discussion does not summarise the findings and contains an unclear description of many imaging techniques. Some examples of the writing style errors and lack of clarity are provided below but are by no means details of every instance. The paper should be completely rewritten in clear simple academic scientific English with the aims and objectives of the study made clear and explicit.

We thank the reviewer for the remark that “The paper contains high quality data and figures and may be of interest to the neuroimaging community.” We agree that the writing style had to be strongly improved, and have completely rewritten the manuscript in consultation with a professional editor who is a native English speaker.

Abstract

Which species?

“Here we demonstrate that neurofunctional ultrasound can exploit its millisecond time resolution within a few hundreds of microns to establish the dynamic role of brain region activations.”

This is not really a sentence. Also, contradicts itself a “few hundred microns”, then claims to be able to image deeper layers?

We agree. Our technique is able to image much deeper than a few hundreds of microns. We have rewritten and clarified the sentence

Peculiar use of the term avant garde ?

“This paves the way to some avant-garde live imaging developments and create novel perspective for deeper understanding of brain network dynamics.”

Corrected

Intro

Line 6 – size of what?

14 valuable is a value judgement, they are all valuable - depends what you are trying to do?

23 – possibly true but what about event related fMRI

36 instantly capture – what does this mean?

11 experimental framework, what is a framework, better to write in more simple clear terms.

We corrected this style error

“Nicely fluctuate” not typical academic style

We corrected this sentence and made our best efforts to delete non-scientific wording in the revised manuscript.

Discussion

The first paragraph is unclear, which technique are we using here? Which technique are you comparing it to? And which has 5 or 50% sensitivity

We have completely rewritten the discussion.

“Functional data were acquired with 2 trained captive-born macaques (Maccaca 18 mulatta) performing in a row with baseline (rest phase),”

Were the 2 monkeys sat in a row as (as this sentence implies) or were the cognitive tasks performed sequentially?

Reviewer #2 (Remarks to the Author):

This study reports on a ultrafast Doppler-based imaging approach (fUS) for measuring fluctuations in cerebral blood volume (CBV) with a high spatial (100um)

and temporal (10ms) resolution in two well-separated brain areas (SEF and ACC) of two awake, behaving non-human primates. While this is without a doubt a technical 'tour de force', I'm not sure what the main contribution is and question the biological relevance of the acquired data.

The aim of our study was primarily to demonstrate the ability of functional ultrasound to distinguish brain activity in the non human primate performing a cognitive task. We have shown that the sensitivity, spatiotemporal resolution, and ease of use of fUS means that it is a valuable new tool for behavioral studies when compared to fMRI and electrophysiological techniques.

We do, however, also consider that this study investigated a biologically relevant question. As reviewer #2 certainly knows, both the ACC and SEF have many functions, including proactive and reactive cognitive control (see Alexander & Brown, 2010 for a review). However, there is still debate as to the specific role of the ACC and SEF in the monitoring of action and cognitive control.

One reason why this debate has been hitherto difficult to resolve empirically is that event related potentials (ERP) and functional magnetic resonance imaging (fMRI) cannot offer sufficient spatial or temporal resolution to do so. Combining simultaneous extracellular recordings in large deep areas is technically challenging. Ultrafast ultrasound imaging offers the prospect of resolving these alternative hypotheses by simultaneously recording CBV in large areas of interest with a high temporal resolution.

Our results provide clear support for the models that predict that the ACC is activated earlier during preparatory periods (pro-active control), whereas the MFC, and the SEF in particular, are activated later and monitor for conflict during stimulus processing and response selection.

We have shown that the fluctuations in CBV recorded by fUS during oculomotor control tasks are temporally synchronized with the individual trials (Fig. 2) and demonstrate that this level of synchronization is correlated with and even predictive of the success rate. Finally, we indicate the ability of neuro-functional ultrasound to provide monitoring of the dynamic propagation of endogeneous signals through cortical layers and between SEF and ACC. This technique is unprecedented (Fig. 3).

Major:

1- The authors make the point that fast imaging of CBV may help making direction inference of neural communication. However, CBV changes must be met with changes in vessel size (i.e. dilation/constriction) which typically takes place on the

order of seconds (not ms, like neuronal activity). To me, Figures 1c, 3 and 6 seem to show this: Area AC responds ~1-2s before area SEF. This is well-beyond neuronal communication timescale, and more likely represents conducted or retrograde dilation within blood vessels. That is, dilation-mediated increases in CBV are first manifested in deep paryenchymal vessels (area ACC?) and then back propagate towards the larger surface vessels (area SEF?, also see O'Herron et al., Nature 2016). Should this be the case, it would suggest that they are not measuring neuronal communication between areas per se (i.e. a top-down neuronal effect), but rather simply providing further evidence for vascular communication, thus questioning the novelty of the current study. The authors need to address this.

We thank Reviewer 2 for this important and interesting comment. We do not think that dilation-mediated increases in CBV can be the origin of our directional connectivity in our experiments. We have presented further data (on spontaneous activity) in the results section, and completely rewritten the discussion to address this concern. (The four paragraphs beginning "It could be argued in favour of a purely vascular effect due to conducted or retrograde dilation within blood vessels (O'Herron et al., Nature 2016)...)

2- I struggled with the meaning and significance of the 'behavioral signal'. A schematic showing the temporal evolution of a successful vs unsuccessful trial would have been helpful. As a result, I could not appreciate the relevance of the CBV-behaviour correlations.

We apologize for this lack of clarity. We have revised the figure and amended the description in the methods.

Minor:

1. How was area AC defined?

Taking into account the fact that the chamber was inserted using precise stereotactic coordinates, SEF and ACC structures were defined based on the sulcus position on vascular maps and neuroanatomical knowledge of the area.

2. Cardiac and respiratory artefacts should be validated with additional measures.

This point was addressed and validated in detail in Osmanski et al Nature Comm 2014. This is a well-known feature of fUS imaging. We have added a citation and expanded the discussion to address this point.

3. Figure 2a is impressive, though I'm surprised the authors did not exploit the data further by examining the spatial extent of this effect (rather than using a 240 pixel ROI, which seemed arbitrary to me).

We thank reviewer 3 for this positive comment. As our objective was to study the SEF, we defined our ROI according to the SEF spatial extent (visible in figure 1).

4. The paragraph in the discussion describing the many limitations of BOLD/ASL is long and doesn't really add much since these data were not acquired in the current study, making it impossible to quantify the true SNR advantage of fUS. In my opinion, this would have substantially increased the importance of this work.

We deleted the paragraph on BOLD signal. Although the comparison with fMRI sensitivity would be interesting to provide quantitative values, there is a very large difference between the signal-to-noise ratios of BOLD and fUS. The rewritten manuscript summarizes this point by stating that the fUS signal is typically ten times larger (50% increase compared to baseline) than noise fluctuations (5% standard deviation of baseline).

5. While it's true that BOLD may not be able to accurately measure rapid signal transmission across the laminae, BOLD is in part sensitive to O₂ metabolism, which is arguably a closer correlate to neuronal activity than CBF and CBV. In this sense, it could be argued that fUS suffers from inherent limitations (i.e. vessel dilations which are slow) thus making it no better than BOLD. The authors should comment on this.

It is completely true that fUS imaging is intrinsically limited by neurovascular coupling as is fMRI. However, fUS's very high temporal resolution makes it possible to estimate very small latency delays between two separated spatial points when measuring changes in CBV. This is important, as it is possible to access to fast propagative information between two points even if the signal being studied is slowly varying. In wave physics, this is the difference between the local particle velocity and the propagation speed of these local particle velocity changes. In other words, even if the local signal is slowly varying in time, imaging its changes in space and time at higher space and time resolutions enables its fast propagation to be tracked. This is why the order of magnitude increase in temporal resolution in fUS (10 ms, compared to fMRI's 200 ms) combined with 100 μ m spatial resolution gives access to propagation of activity even in the case of spontaneous activity where propagation is fast (around 0.4 m/s). Our results are in agreement with previous studies (around 0.2 m/s) using VSD calcium imaging in mice (Mohajerani et al. 2010, 2013).

Reviewer #3 (Remarks to the Author):

This paper examines the measurement of cerebral blood volume changes with function ultrasound (fUS) in response to visual cues in macaque primates. The blood flow in the supplementary eye field (SEF) and anterior cingulate cortex (ACC) was measured after different cues (fixation, saccade, and anti-saccade). The paper has interesting data, but needs to clarify several aspects. The temporal resolution of the fUS method is a strong advantage of this paper for resolving aspects of responses in the brain to visual cues.

1. There are a fair number of superlatives that appear throughout the text (Auspiciously) that may need to be toned down.

We are sorry for this. We fully agree and toned down our excessive enthusiasm. We made our best efforts to delete non-scientific wording in the revised manuscript.

2. What is the role or point of the 1 pixel region-of-interest? This is not properly motivated and honestly shows much larger variation than using a larger area.

We wanted to show that the CBV changes were already detectable within a single pixel although spatial averaging is of course increasing the raw data quality. A comment was added to better introduce the reason for 1pixel ROI.

3. This is expected, but the justification of this alternative analysis is missing.

This alternative analysis was devoted to show that results can be robustly obtained even within a single pixel.

4. Figure 2. What defines trial success? This is a fundamental component that was not properly defined.

Thanks for this comment too. You are right and we have better defined the trial success.

5. How was the seed for the seed-based correlation analysis chosen? Was the seed varied and did it affect the results?

The seed was chosen within the SEF on the most correlated pixel with the task. Changing the seed within the SEF did not affect the results in a meaningful way. As a propagation is clearly visible in the supplementary video, the goal of this analysis was mainly to provide estimates on the propagation speed to be compared with the literature.

6. In the SEF region the propagation is top-to-down, but in the ACC it is down-to-top. This needs to be reflected in the analysis and description.

We are sorry but we feel there is a misunderstanding of the classical “top-down” and “bottom-up” terminology in Neuroscience. Top-down describes

the propagation from top layers to deeper layers. Bottom-up describes the propagation from deep layers to top layers. As the cortex is following the curvature of the gyrus (figure 3b), propagation is top-down in the SEF but is also top-down in the ACC. In the ACC, the activation begins in the top layer and moves to the deeper layers. This strongly weakens the hypothesis of a purely vascular origin of the observed directional connectivity. In order to clarify this points, we added top and down labels in the supplementary figure as well as layers contours.

7. The invasiveness of this technique needs to be addressed as a limitation in the Discussion.
You are right and we added comments on this limitation in the discussion.
8. The ethical approval for animal testing needs to be addressed in the Methods.
Done in the corrected version
9. Figure 2d. What is the implication for the correlation for the first 40 seconds to later measurements?
The degree of synchronization during the first 40 seconds i.e. during fixation block, was found to predict the performance of the subsequent trials during saccade and antisaccade blocks. This synchronization during fixation could thus reveal a degree of attention of the animal for a particular session.
10. Figure 3b. The points are very sparsely distributed which makes the results somewhat suspect with respect to the robustness. The authors need to address this. Would longer averaging times in slow time be useful to make more robust measurements?
*In order to select only robust pixels for the analysis, we applied a threshold on the correlation value of the pixels (>0.3). This ensures that only robust pixels are used in the time analysis. It would be possible to use a lower correlation threshold but that would reduce overall robustness of the time estimation. One should note as well that since the seed or reference signal is in the SEF, the correlation with the ACC pixels is lower since they are much farther away.
*Averaging could indeed increase individual pixel correlation by reducing electronic or physiological noise or activity unrelated to the task and thus increase the number of available pixels in the analysis.**
11. Figure 3c. The Roman numerals on the right of the propagation plots need to be defined.
Done in the corrected version
12. Figure 4. The “SC ?” need to remove the question marks.

Sorry for this. It is done in the corrected version.

12. Figure 6. Again, these correlation maps are incredibly sparse. How was the path of propagation (the spatial axis) defined? Was it done on a Cartesian grid as shown in the image or some other path? The path needs to be shown on the images. Also, in the estimation of the propagation the feature being tracked needs to be stated. There is a good deal of variation in the propagation plots and the R^2 values of the fits should be reported to provide a confidence metric for the reader.

The path of propagation was defined manually along a line perpendicular to the layers in both the ACC and SEF and is represented as white arrows. Layers are now drawn on top of the image to better visualize the anatomy. There is indeed some jitter in the propagation plots but note that there is a clear propagation that is also visible in the supplementary video. Again the goal of this analysis was to provide estimates of the propagation speed to compare with the literature. We also added propagation speed estimates on spontaneous activity in the revised version.

REVIEWERS' COMMENTS:

Reviewer #1 (Remarks to the Author):

The report has been greatly improved after addressing all the comments from the initial reviewers. It is now clearly written and it is articulated how it is of interest both in terms of the development of neuroimaging techniques and cognitive neuroscience. It is suitable for publication without further revision.

Reviewer #2 (Remarks to the Author):

The authors have fully addressed my major concerns. I think the presentation of the results has substantially improved, and is now in a publishable form.

Reviewer #3 (Remarks to the Author):

The revised version of this paper is greatly improved. Some of my previous concerns were addressed in the rebuttal but not satisfactorily handled in the text.

1. I still question the relevance and validity of the 1 pixel region-of-interest (ROI). There is significantly more variation in the 1 pixel ROI which is not surprising. I would trust a region of perhaps 9 pixels or something on that order. I question if that pixel is consistent in space due to cardiac pulsations and other physiologic motion. Fig. 1(b) shows cardiac pulsatility but there are no units. What was the level of axial and lateral motion related to the pulsatility?
2. The limitations noted in the paper are few. The authors could have scrambled the order of the fixation, saccade, and anti-saccade to make the task more complex. Also, variation of the time between trials could have been varied to make sure that the frequency content was representative of the tasks being analyzed. These would have been good tests to evaluate the robustness of the method and check its validity in different conditions.
3. Page 11. Could averaging be used to improve the correlation maps? I still maintain that they are fairly sparse and raise some concerns about validity due to that sparseness. This is especially true in the ACC.
4. I could not find any text related to the ethical approval for the animal use in these experiments.
5. Fig. 3(b). Why does the time-domain correlation have more variability than the frequency domain? That is, there is more spread in the time-domain data with respect to the correlation line.
6. Figure 7. The SC? labels still persist in the 15 and 12 MHz images.

The revised version of this paper is greatly improved. Some of my previous concerns were addressed in the rebuttal but not satisfactorily handled in the text.

1. I still question the relevance and validity of the 1 pixel region-of-interest (ROI). There is significantly more variation in the 1 pixel ROI which is not surprising. I would trust a region of perhaps 9 pixels or something on that order. I question if that pixel is consistent in space due to cardiac pulsations and other physiologic motion. Fig. 1(b) shows cardiac pulsatility but there are no units. What was the level of axial and lateral motion related to the pulsatility?

We agree with the reviewer. Assessing activity of a brain region using a single pixel is challenging. fUS signal was assessed in several size of ROI in the SEF, for more clarity we kept only the large ROI and the 1-pixel ROI. Indeed, as shown below (Figure 2b enhanced, non-submitted in this form) we have tested sensitivity of fUS considering only one trial for ROIs of 1, 6, 12 pixels and large ROI (~200 pixels) with the specific condition that the 12-pixels ROI contains the 6-pixels ROI which contains the 1 pixel-ROI.

Even if there was more variability in the signal in the 1-pixel ROI it was sensitive enough to reveal single trial (Figure 2b). The following sentence was added to clarify Page 8, line 13-14: (similar results were obtained in ROI of 6 and 12 pixels, but for more clarity data were not shown).

Moreover, considering the similarity (figure below) of the signal in 1 pixel, 6 pixels and 12 pixels, we considered that the information contained in only 1 pixel was reliable to infer on the neural state of the subject.

In figure 1b is presented the spectrum of CBV in SEF so the x axis is in Hz, the cardiac pulse activity indicated is around 3.5 Hz. As described in figure caption this is equivalent to 210 beats per minute (60sec*3.5Hz).

Pulsatility does not seem to involve tissue displacement greater than 100 μ m as we don't see any movement on doppler movie. There might be some movement, but it should happen at sub-resolution pixel. indeed, pulsatility is strongly canceled by the spatiotemporal clutter filter as demonstrated by Dememe et al (Demene et al, Spatiotemporal clutter filtering of ultrafast ultrasound data highly increases Doppler and fUltrasound sensitivity. IEEE Trans Med Imaging 2015).

2. The limitations noted in the paper are few. The authors could have scrambled the order of the fixation, saccade, and anti-saccade to make the task more complex. Also, variation of the time between trials could have been varied to make sure that the frequency content was representative of the tasks being analyzed. These would have been good tests to evaluate the robustness of the method and check its validity in different conditions.

Again, we totally agree and this is what will perform in future protocol. Several acquisitions were made performing only baseline followed by antisaccade, 1 example is presented in supplementary material figure 4d (there is a mistake in figure number increment, it should be figure 5) the spectrogram revealed CBV oscillations around a frequency of \sim 0.30Hz which is typical frequency related to response time of antisaccade as shown in figure 3b (frequency domain). In future work with new primates, we will invert the order of visual tasks and we will increase the jitter between two start-trial.

We have added the following paragraph in the discussion section, page 11, line 1-11: As presented in the spectrograms of figure 3a, a significant shift of the CBV oscillations frequency was found when changing rule of visual task. We verified the hypothesis (fig. 3a and 3b) that the shift was directly related to the RT of animals by comparing the peak frequency of CBV signal spectra in SEF with the behavior signal spectra for each type of visual task (see material and method and fig. 1c for detailed descriptions about behavior signal). It could be argued that this shift in frequency of the CBV signal in SEF could be related to fatigue of the animal instead of its response time. But as presented in supplementary figure 4d, we have performed several acquisitions with only baseline followed directly by antisaccade and we saw that CBV oscillations frequency were around 0.3Hz for the whole acquisition which is typical frequency related to RT for antisaccade (fig. 3b). In future work, with new primates, we will invert the order of visual tasks and we will increase the jitter between two start-trial.

3. Could averaging be used to improve the correlation maps? I still maintain that they are fairly sparse and raise some concerns about validity due to that sparseness. This is especially true in the ACC.

This is an interesting point. When we have put a seed in the pixel with the highest CBV variation in the SEF we expected to find only correlations around the seed pixel, and we were surprised to find correlation in farther region such as dorsal and ventral ACC. We are averaging temporally as we are taking around 20 trials (\sim 60 seconds) to assess correlations between seed-pixel and other pixels of the image. If the reviewer is referring to spatial averaging, if we do so we would lose in spatial resolution and one interest of our method is in future to target specific pixel with electrophysiology electrodes.

The following sentence was added line 5-7 page 15: Raw spatial correlation map presented in figure 4b demonstrates pixel-level activation in ACC. Although an increase of spatial filter could be applied to further improve the smoothing of activation maps, preserving high spatial resolution could be crucial for several applications in particular when a precise guiding of electrodes is required doing electrophysiology measurements.

4. I could not find any text related to the ethical approval for the animal use in these experiments.

We thank the reviewer for that mistake. We have added ethical approval reference line 21-23 page 5: All experiments were ethically approved by the French "Ministère de l'Éducation, de l'Enseignement Supérieur et de la Recherche" under the project references APAFIS #561_2015042717569705 and #6355-2016080911065046.

5. Fig. 3(b). Why does the time-domain correlation have more variability than the frequency domain? That is, there is more spread in the time-domain data with respect to the correlation line.

Once again, we thank the reviewer for his/her comments. Time-domain correlation have more variability certainly because the period between CBV peak was assessed by performing automatic detection of maxima whereas the peak in frequency domain was assessed by using the fast Fourier transform of the temporal signal which seems to be more robust to define the periodicity of a quasi-periodic signal.

6. Figure 7. The SC? labels still persist in the 15 and 12 MHz images.

Good catch. We have corrected that.